# Paravalvular Leak Echo Imaging before and during the Percutaneous Procedure

**DOI:** 10.3390/jcm11113155

**Published:** 2022-06-01

**Authors:** Piotr Pysz, Wojtek Wojakowski, Grzegorz Smolka

**Affiliations:** Department of Cardiology and Structural Heart Diseases, Medical University of Silesia, 40-635 Katowice, Poland; wwojakowski@sum.edu.pl (W.W.); gsmolka@me.com (G.S.)

**Keywords:** paravalvular leak, device closure, echocardiography

## Abstract

Percutaneous device closure has become a valuable alternative to surgery in the management of paravalvular leaks. Consequently, imaging in these patients is currently not only meant to verify the hemodynamic significance of the lesion but also to assess the feasibility of transcatheter treatment. We present a methodology of comprehensive echocardiography assessment that allows for the selection of patients and plans the intervention. Next, procedure-oriented steps of echocardiography imaging, which are essential for eventual success, are reviewed.

According to current guidelines [1,2], the repair of paravalvular leak (PVL) should be considered in patients with heart failure symptoms and/or severe hemolytic anemia. Patients at high surgical risk and with a favorable leak morphology are candidates for catheter-based therapy performed in experienced valve centers.

In this context, except for laboratory diagnoses of severe hemolytic anemia, the qualification of patients for device closure is predominantly based on imaging. As reliable exclusion of chronic subclinical endocarditis smoldering in the tissues surrounding the PVL is difficult, the postponing of the transcatheter procedure for at least 6 months from the last clinically overt IE episode seems to be a safe approach. In patients with clinical indications for more urgent intervention, positron emission tomography (PET) may be helpful.

Baseline echocardiography should assess the hemodynamic significance of the PVL and the feasibility of its percutaneous repair. A comprehensive study is mandatory, including transthoracic (TTE) and transesophageal (TOE) examination with 2D and real-time three-dimensional (RT-3D) imaging.

Grading the severity of PVL has now been standardized for both surgical and transcatheter prostheses [3,4,5]. It should be based on an analysis of multiple structural and doppler ultrasound parameters. First, the sewing ring motion is evaluated to exclude the instability of the prosthesis. Typically, “rocking” accompanies severe PVL caused by large dehiscence, usually ≥40% of the sewing ring. Sometimes, hypermobility of the prosthetic valve may also be due to increased pliability of the adjacent tissue, more frequently in the mitral position, but also in cases of post-endocarditis lesions surrounding the aortic prosthetic valve. In these cases, neither prosthesis instability nor a PVL presence is necessarily implied. In the next step, the doppler parameters need to be inspected, as listed in Table 1 and Table 2.

From a practical standpoint, a semiquantitative assessment of flow pattern distortion with pulse wave doppler upstream (pulmonary veins for mitral PVL) or downstream (descending aorta for aortic PVL) from the valve seems particularly useful, as it is both easy to obtain in all patients in TOE and hardly affected by an observer-related error in measurement. Second, an elevated transprosthetic gradient (flow) despite normal appearance and mobility of the cusps/disk (s) should raise a suspicion of significant PVL causing the observed volume overload. Finally, a lack of LV dimensions’ normalization following baseline valvular intervention or progressive LV enlargement may also be suggestive of severe PVL. On the other hand, it should also be underscored that the selected population of patients may benefit from early intervention, even when the PVL is judged to be only mild/moderate (i.e., post-TAVR patients with small, hypertrophied, and rigid LV and no preexisting aortic regurgitation) [6].

Once the hemodynamic indications for intervention are confirmed, the echo study should focus on preparation for the percutaneous procedure, which includes choosing the access site, the delivery system, and the occluder device(s). For these, the number of PVLs, their location, their shape, and their size are relevant. Ongoing development of plugs delivery techniques presently enables closure of the vast majority of PVLs in a fully percutaneous, truly minimally invasive manner (transvenous antegrade for the mitral or tricuspid ones and transarterial for the aortic ones). For aortic PVL, it is usually sufficient to describe its location by relating to coronary sinuses or native cusps/commissures. Such a description allows for an adequate choice of coronary catheters used to reach and cross the PVL. The mitral PVL site, on the other hand, should ideally be referred to the surgical clock, with 12:00 o’clock marked by the aorta and 3:00 o’clock by the interatrial septum. Such detailed localization facilitates the optimal selection of the transseptal puncture site (see the section titled “Echo imaging during the procedure”). Regarding the shape and size of the PVL, an RT-3D TOE is exceptionally informative. The algorithm of image acquisition and analysis has been developed in the PVL-reference center represented by the authors and reported previously [7]. In brief, the smallest possible volume containing the complete PVL channel in an ECG-gated, single-beat zoom 3D high volume rate (HVR) mode with color doppler (CD) is acquired. Next, PVL opacification by CD-mapped flow turbulence is analyzed in multiplanar presentation. This enables the visualization of the PVL’s vena contracta shape and measuring its dimensions (minimal and maximal extent, area, and length—Figure 1A,B and Appendix A) with high reproducibility.

Based on this analysis, a choice can be made between the two devices currently registered for transcatheter PVL closure in the EU and the USA. For regular (oval/round) and short channels (<5 mm) with no structures (e.g., calcifications) preventing apposition of discs, a paravalvular leak device (PLD, Occlutech) is selected. For the irregular ones with long channel (>5 mm), calcifications and/or prosthesis horns in the vicinity of PVL Amplatzer Valvular Plugs 3 (AVP3, Abbott) implanted in a multiplug technique appear to be a better option. In both scenarios, high safety and efficacy have been reported [8,9]. Fairly recently, new 3D image rendering techniques have been introduced to allow, first, the visualization of the morphology of the PVL channel in an even more detailed way, and later, to efficiently guide the procedure [10]. Importantly, the septa (sutures/calcifications) dividing the area occupied by PVL into subcompartments can be either confirmed or ruled out, which influences the choice of the occluders. See Figure 2 and Appendix A. In infrequent cases of acoustic shadowing, disabling sufficient visualization of the PVL channel despite attempting multiple views at different depths of probe introduction, supplementary data and measurements may be obtained by CT.

Intraprocedural echo imaging is crucial for all steps of transcatheter PVL closure. It supports optimal transseptal or transapical puncture, facilitates finding and crossing the PVL channel, and visualizes the level of deployment of the plugs, particularly in cases of prosthetic valves that are poorly visible on fluoroscopy. In addition, it helps avoid complications (tamponade, prosthetic valve impingement, and chordae entanglement). In most cases, TOE is the modality of choice [11,12], but the use of intracardiac echocardiography (ICE) has also been reported [13,14]; is predominantly advocated to avoid TOE-related general anesthesia (GA). Nevertheless, it should be emphasized that various conscious sedation protocols, chiefly based on the concomitant intravenous administration of benzodiazepine and fentanyl, enable nearly all procedures to be performed under TOE guidance and without GA. Furthermore, ICE occupies one of the operators maneuvering the probe, while offering a relatively narrow angle of scanning with three-dimensional imaging quality inferior to TOE. In rare cases of patients with contraindications to TOE, a TTE monitoring may substitute TOE during aortic PVL closure, at the expense of an increased use of contrast agent and radiation dose, also for the echocardiographer.

For mitral PVLs, the choice of a transseptal puncture site should be driven by the location of the channel, as such an approach greatly enhances the chances of successful crossing. Ideally, echo guidance should offer biplane imaging, with two perpendicular planes simultaneously visible side by side (without the need for repetitive switching between bicaval and short-axis views). While tenting produced by the transseptal sheath is visualized in the center of both images, its location can be further optimized. See Figure 3.

The directions of motion are identified in a standard manner in regard to the superior (SVC) and inferior (IVC) vena cava (superior: towards the SVC; inferior: towards the IVC) and aorta (anterior: toward the aorta; posterior: opposite the aorta). Increasingly, in selected patients, intuitive guidance by 3D volume rendering is possible. See Figure 4.

For septal and lateral located PVLs, a central puncture is usually optimal, given the height above the mitral annulus sufficient to accommodate the curvature of the steerable sheath (depending on model and size of the sheath, usually ca 2.5 cm). For anteriorly situated PVL, a more superior puncture may be better, as it usually improves the axiality during the crossing. The posteriorly located PVL remains the most challenging for transseptal access. Usually, a low and posterior puncture is necessary to achieve axiality enabling the crossing. However, this may be limited by the individual anatomic relations between the size of fossa ovalis and the angle at which the IVC enters the right atrium.

Occasionally, in patients with posteriorly located PVLs and anatomies “hostile” for an optimal transseptal puncture, a retrograde, transapical, or a transarterial approach, may prove to be more feasible. Again, during the transapical puncture performed with lateral mini-thoracotomy, TOE may help to identify the true apex of the left ventricle while pressure is locally applied from outside (e.g., by the operator’s finger or by surgical forceps). Thus, puncture of the distal segments of the right ventricle and/or intraventricular septum is avoided.

Once the transseptal puncture is accomplished, echo imaging greatly simplifies the maneuvering towards the PVL site with the predominant role of 3D TOE, which has already been specifically addressed in this edition of Special Issue [15]. Still, especially in the case of a narrow PVL channel, finding and crossing may be difficult. In such cases, once the tip of the catheter is placed in close vicinity to the PVL orifice as assessed by 3D TOE, it may be worthwhile to switch back to biplanar imaging with color doppler marking the PVL. See Figure 5A,B.

This allows for steering the gentle movements of the catheter/wire towards the PVL, simultaneously in two directions, with a temporal and spatial resolution of the image superior to 3D. For aortic PVLs, it is mostly the 2D TOE short-axis views that are used to identify the position of the tip of the catheter in the aortic sinus, with 3D TOE offering sufficient quality of image only in selected patients.

After crossing the PVL and advancing the delivery catheter with plug(s) echo monitors the position of the devices. Typically, during retrograde closure of aortic PVLs, but sometimes also during antegrade closure of mitral PVLs (patients in whom the subvalvular apparatus was largely preserved during surgery), the partially open occluders, while being retracted towards the sewing ring, may become entangled in the chordae. This is easily identified by TOE and leads to device repositioning. Next, echo imaging, while visualizing the true sewing ring (as opposed to only its X-ray opaque parts visible on fluoroscopy), helps to control the level of deployment of the occluders and is even more crucial in patients with prosthetic valves poorly visualized or invisible by X-ray (especially stentless valves and some stented bioprostheses). See Figure 6.

At this point, biplanar views may simultaneously visualize the position of the plug and ensure the unaffected mobility of the prosthetic discs. See Figure 7A,B.

Once the occluder(s) are fully opened in the PVL but still connected to the delivery cable, echo focuses on assessing the acute result. This involves CD examination with measurements of residual PVL if present (VC, VC area by 3D TOE) and doppler parameters (transprosthetic gradient, flow pattern in pulmonary veins or descending aorta, depending on the PVL location). Obviously, all values are compared to those at baseline (which should be recorded not only during the qualification of the patient, but also mandatorily at the beginning of the procedure, and in hemodynamic conditions possibly comparable to these at the end). Importantly, it has been reported for the mitral PVLs that postprocedural ≤ mild regurgitation on echo corresponds to improvement in invasively determined hemodynamic parameters [16]. A residual regurgitation should not be accepted, regardless of its severity, if the flow is present across the plug, as it increases the risk of postprocedural exacerbation of hemolysis [17]. Persistent flow around the plug may be accepted proven trace or mild on CD with normalization of the above-mentioned doppler parameters. Mild-to-moderate or larger residual PVL should lead to plug repositioning or replacement. It is also possible to plan a staged procedure with implantation of additional devices, provided a stable position on a tug-test is achieved. In general, complete elimination of PVL should be achieved, as insufficient reduction of regurgitation adversely influences the long-term outcome [18].

## Figures and Tables

**Figure 1 jcm-11-03155-f001:**
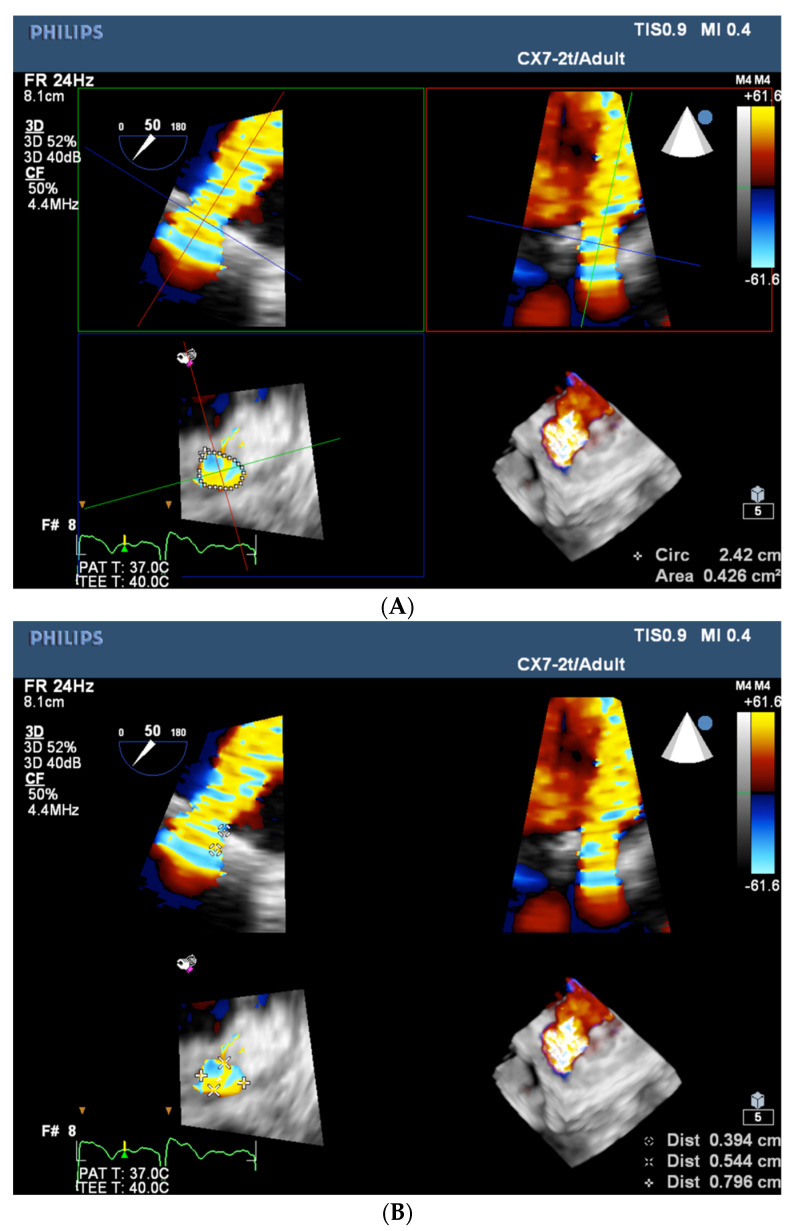
(**A**) RT-3D TOE multi-planar visualization of PVL’s vena contracta—measurement of its area (left lower panel). (**B**) RT-3D TOE multi-planar visualization of PVL’s vena contracta—measurement of its length (left upper panel) along with minimal and maximal extent (left lower panel).

**Figure 2 jcm-11-03155-f002:**
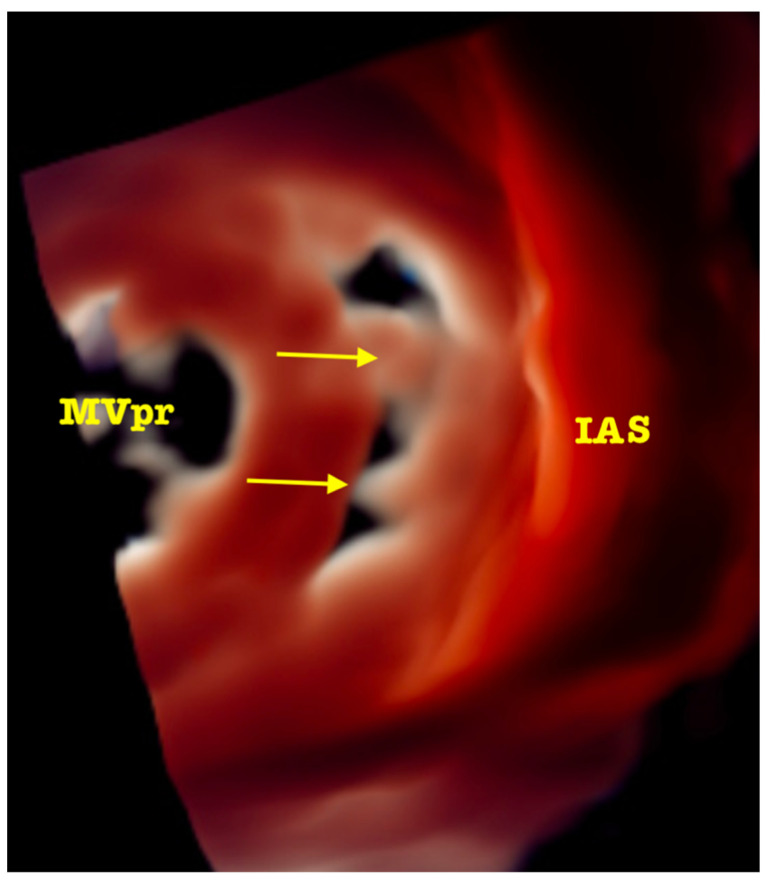
Advanced 3D image rendering with artificial “source of light” placed in the LV cavity allowing the precise visualization of the septa (arrows) dividing the PVL lumen (MVpr—mitral bioprosthesis, IAS—interatrial septum).

**Figure 3 jcm-11-03155-f003:**
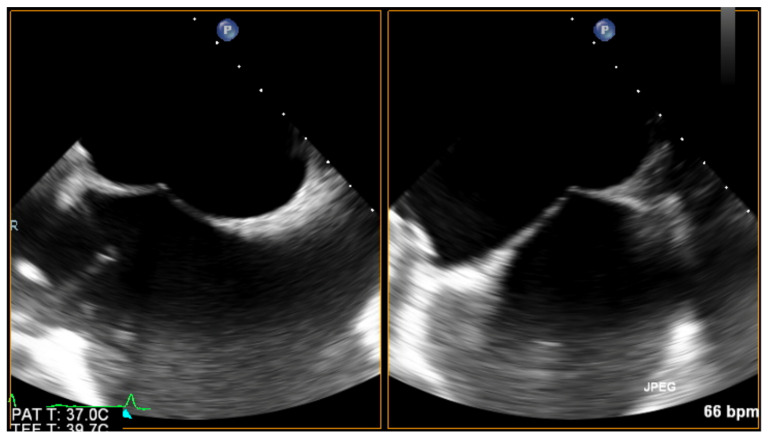
Biplanar imaging of the interatrial septum tenting produced by the tip of the transseptal sheath.

**Figure 4 jcm-11-03155-f004:**
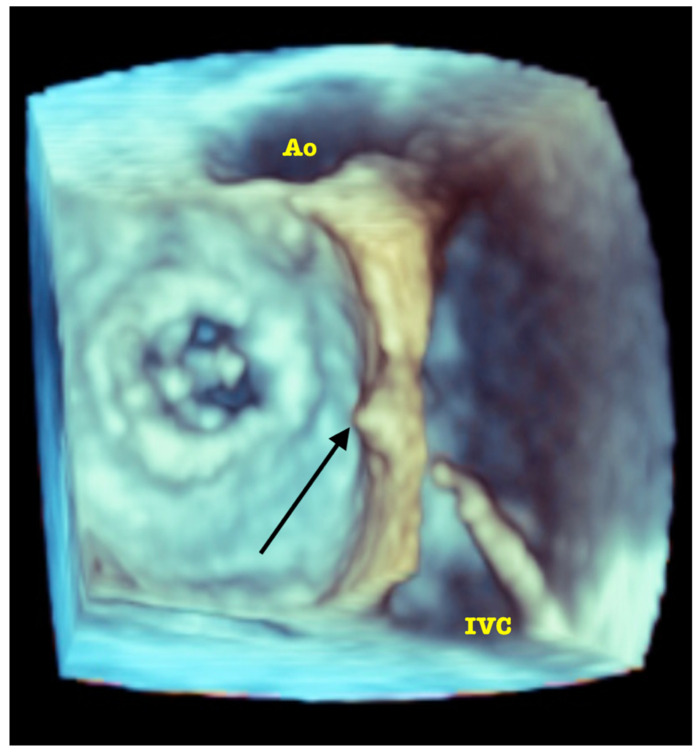
3D TOE volume rendering visualization of the interatrial septum tenting (black arrow) produced by the tip of transseptal sheath (IVC: inferior vena cava; Ao: aortic root).

**Figure 5 jcm-11-03155-f005:**
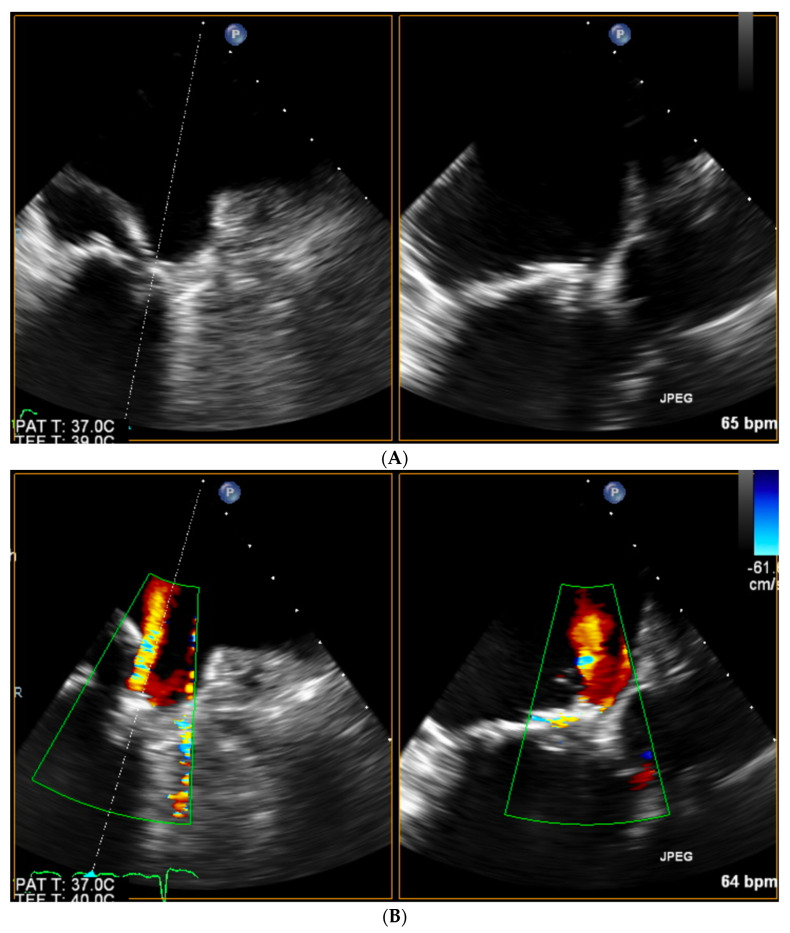
(**A**) Biplanar imaging of the tip of the catheter at the sewing ring of prosthetic valve. (**B**) Biplanar color doppler marking of the PVL’s narrow orifice enabling precise catheter maneuvering.

**Figure 6 jcm-11-03155-f006:**
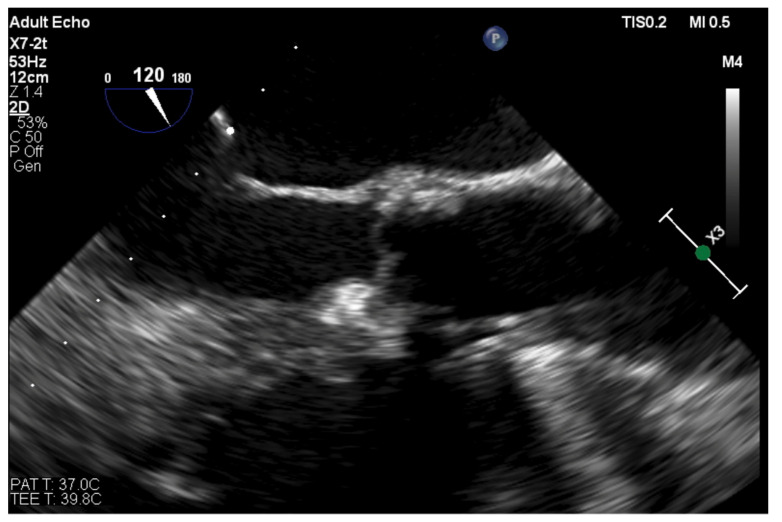
TOE long-axis view of the aortic bioprosthesis poorly visible on X-ray—distal disc of the occluder visualized while being implanted in the PVL located at the right coronary sinus.

**Figure 7 jcm-11-03155-f007:**
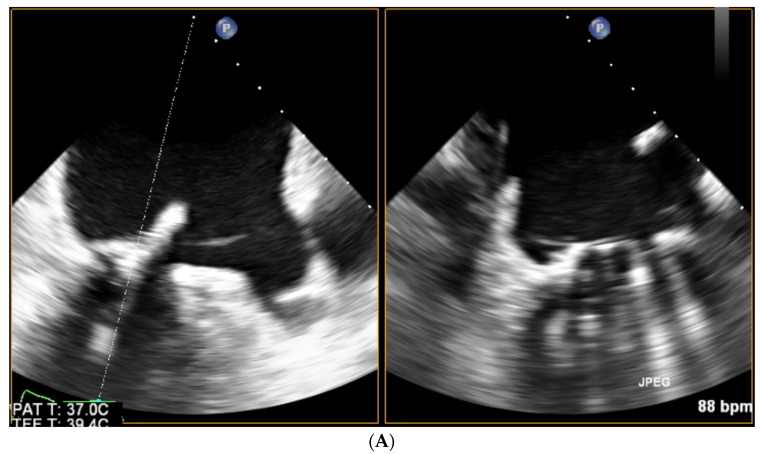
(**A**) Mitral PVL closure, TOE biplanar imaging allowing for simultaneous controlling of the plug implantation (left panel) and preserved prosthetic discs opening (right panel—diastole, discs symmetrically opened). (**B**) Mitral PVL closure, TOE biplanar imaging allowing for simultaneous controlling of the plug implantation (left panel) and preserved prosthetic discs closing (right panel—systole, discs symmetrically closed).

**Table 1 jcm-11-03155-t001:** Selected echocardiography parameters useful for aortic PVL grading [1,2,3,4].

	Trace	Mild	Mild/Moderate	Moderate	Moderate/Severe	Severe
Flow convergence	Absent	Absent	Absent	Possible	Usually present	Usually present
VC, mm	Untraceable	<2	2–4	4–5	5–6	≥6
PHT, ms	>500	>500	200–500	200–500	200–500	<200
Diastolic flow reversal in descending aorta	Absent	Absent/early diastolic	Intermediary	Intermediary	Holodiastolic (EDV 20–30 cm/s)	Holodiastolic (EDV ≥ 30 cm/s)
Circumferential extent of PVL, %	Untraceable	<5	5–10	10–20	20–30	≥30
RVol, mL	<10	<15	15–30	30–45	45–60	≥60
RF, %	<15	<15	15–30	30–40	40–50	≥50
EROA, mm^2^	<5	<5	5–10	10–20	20–30	≥30

VC—vena contracta, PHT—pressure half-time, EDV—end-diastolic velocity, RVol—regurgitant volume, RF—regurgitant fraction, EROA—effective regurgitant orifice area.

**Table 2 jcm-11-03155-t002:** Selected echocardiography parameters useful for mitral PVL grading [1,2,3,4].

	Trace	Mild	Mild/Moderate	Moderate	Moderate/Severe	Severe
Flow convergence	Absent	Absent/minimal	Absent/minimal	Intermediary	Intermediary	Large
VC, mm	Untraceable	<2	2–3	3–5	5–7	≥7
Circumferential extent of PVL, %	Untraceable	<5	5–10	10–20	20–30	≥30
RVol, mL	<10	<15	15–30	30–45	45–60	≥60
RF%	<15	<15	15–30	30–40	40–50	≥50
EROA mm^2^	<5	<5	5–20	20–30	30–40	≥40

VC—vena contracta, RVol—regurgitant volume, RF—regurgitant fraction, EROA—effective regurgitant orifice area.

## Data Availability

Not applicable.

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
