# Peer review of "Paravalvular Leak Echo Imaging before and during the Percutaneous Procedure"

_jcm, 2022, doi:10.3390/jcm11113155_

Round 1
Reviewer 1 Report
well written paper. I would recommend making some additions regarding imaging, especially adding more 3D images and explaining orientation and optimal imaging angles for various paravalvular leak closure procedures.
For example consider adding optimal way to get 3D imaging of mitral valve paravalvular leaks
Reviewer 2 Report
Dear authors and editorial staffs
This manuscript is about ‘Paravalvular leak echo imaging before and during the percutaneous procedure’ and recently mitral paravalvular leak intervention has been a major and important therapeutic option, which is valuable to analyze for adequate treatment. I have some comments and questions for authors.
- For multiple mitral paravalvular leak, what is your selection recommendation of the device occlusion or reoperation?
- In many cases, device occlusion made a bigger creek between sewing cuff and mitral annulus. How do authors decide a procedure selection for prevention of this hazardous complication?
- For aortic position para leak, this regurgitation might not be a critical complication in many cases, so what is procedural indication for authors` institution?
Reviewer 3 Report
The "viewpoint" paper entitled "Paravalvular leak echo imaging before and during the percutaneous procedure" presents a short nice review of the utility of echocardiography related to transcatheter PVL closure.
Some minor comments may be done:
- A brief mention of the clinical indications for PVL closure, lab test, and other imaging modalities (CT, PET, …). needed may be included in the paper.
- Figure 3 may be better understood if it would be presented from the surgeon´s view (aortic valve at 12 o’clock) and some anatomic landmark labels are included (aortic root, interatrial septum,…).
- According to new echo technologies, a brief mention may be added about new visualization 3D renders (consider: Barreiro-Perez M, Cruz-Gonzalez I, Villa Gil-Ortega M, Guisando Rasco A, Sanchez PL. Photo-Realistic Echocardiography Imaging During Percutaneous Paravalvular Leak Closure. JACC Cardiovasc Interv. 2020;13(20):e185-e187. DOI: 10.1016/j.jcin.2020.08.031.), and fluoro-echo fusion imaging.
